# Nitrification–Autotrophic Denitrification Using Elemental Sulfur as an Electron Donor in a Sequencing Batch Reactor (SBR): Performance and Kinetic Analysis

**Mario Corbalán [1], Cristopher Da Silva [1], Andrea Barahona [1], César Huiliñir [2,\*] and Lorna Guerrero [1,\*]**

[1] Departamento de Ingeniería Química y Ambiental, Universidad Técnica Federico Santa María, Valparaíso 2340000, Chile; andrea.barahona@usm.cl (A.B.)

[2] Green Technology Research Group, Facultad de Ingeniería y Ciencias Aplicadas, Universidad de los Andes, Mons. Álvaro del Portillo 12455, Las Condes, Santiago 7620086, Chile

\* Correspondence: chuilinir@miuandes.cl (C.H.); lorna.guerrero@usm.cl (L.G.)

**Abstract:** Simultaneous nitrification and autotrophic denitrification (SNAD) has received attention as an efficient biological nitrogen removal alternative. However, SNAD using elemental sulfur ($S^0$) has scarcely been studied. Thus, the main objective of this research was to study the behavior of a simultaneous nitrification–autotrophic denitrification operation in a sequential batch reactor (SNAD-SBR) at a lab scale using $S^0$ as an electron donor, including its kinetics. Two-scale reactors were operated at lab scales in cycles for 155 days with an increasing nitrogen loading rate (NLR: 0.0296 to 0.0511 kg $N$-$NH_4^+$/$m^3$/d) at 31 °C. As a result, simultaneous nitrification–autotrophic denitrification using $S^0$ as an electron donor was performed successfully, with nitrification efficiency of 98.63% and denitrification efficiency of 44.9%, with autotrophic denitrification as the limiting phase. The kinetic model adjusted for ammonium-oxidizing bacteria (AOB) was the Monod-type kinetic model ($\mu max$ = 0.791 $d^{-1}$), while, for nitrite-oxidizing bacteria (NOB), the Haldane-type model was employed ($\mu max$ = 0.822 $d^{-1}$). For denitrifying microorganisms, the kinetic model was adjusted by a half order ($k_{1/2v}$ = 0.2054 $mg^{1/2}$/$L^{1/2}$/h). Thus, we concluded that SNAD could be feasible using $S^0$ as an electron donor, with kinetic behavior similar to that of other processes.

**Keywords:** autotrophic denitrification; kinetic parameters; nitrification; SBR





## 1. Introduction

Recently, simultaneous nitrification and autotrophic denitrification (SNAD) has received attention as an efficient biological nitrogen removal alternative due to the small area needed, smaller footprint, lower energy cost and investment, and simplicity of the process [1]. The SNAD process presents a significant advantage over the conventional separated nitrification and denitrification process. First, the SNAD process eliminates the serial operation of two separate tanks and, therefore, requires more straightforward operational procedures. In addition, SNAD reduces the sludge yield by 30% over conventional biological nitrogen removal (SNR) systems [2].

During the nitrification–autotrophic denitrification process, ammonia nitrogen is oxidized to nitrate and nitrite and reduced to elemental nitrogen. In the presence of reduced inorganic sulfur compounds, it can utilize them as electron donors. Nitrification is typically divided into two main stages: "nitritation," which involves the oxidation of ammonium to nitrite ($NH_4^+ \rightarrow NO_2^-$), and "nitrification," where nitrite is further oxidized to nitrate ($NO_2^- \rightarrow NO_3^-$). These stages occur simultaneously and are mediated by microorganisms known as ammonium-oxidizing bacteria (AOB) and nitrite-oxidizing bacteria (NOB), respectively. Then, autotrophic denitrification occurs, an anaerobic process involving the dissimilatory reduction of oxidized nitrogen compounds ($NO_2^-$ and/or

$NO_3^-$) as electron acceptors. In this process, reduced inorganic sulfur compounds, such as elemental sulfur ($S^0$), act as electron donors [3].

In effluents with a low carbon-to-nitrogen ratio (C/N < 5), as is the case with effluents from anaerobic digesters, or even some effluents from industry, such as tanneries or fertilizer plants, autotrophic microorganism-mediated denitrification is an interesting alternative that allows for the simultaneous removal of nitrogen oxides and reduced inorganic sulfur compounds without the need for supplementation with organic matter [4]. Moreover, the use of autotrophic denitrifying microorganisms has advantages over the heterotrophic process: it can use inorganic substances (such as $S^0$ in this case) as electron donors, eliminating potential problems associated with organic waste, and the addition of external organic carbon is not necessary, reducing the costs and operational risks and eliminating the formation of carbon dioxide. Furthermore, autotrophic denitrifying bacteria that oxidize sulfur produce less nitrous oxide ($N_2O$), a greenhouse gas [1].

Combining processes such as nitrification and autotrophic denitrification in a single reactor, where populations coexist and achieve the simultaneity of the processes, could lead to lower investment and operating costs. This is because the need for a more cost-effective configuration of wastewater treatment plants constantly presents new challenges, such as reducing residual sludge, minimizing the carbon footprint, and decreasing the volumes of the reactors used [5,6].

It is worth noting that the literature reports on nitrification and autotrophic denitrification; however, studies on this topic are needed that delve into autotrophic denitrification carried out in a single reactor simultaneously with nitrification and using elemental sulfur as an electron donor. This represents a novelty in the research of such processes.

Considering these sequential characteristics of the SNAD process, the sequential batch reactor (SBR) is an attractive choice. The SBR is a good option and viable alternative, especially for small- to medium-scale industrial units. Excellent results for pollutant removal have been observed using this technology in advanced nations compared to continuous flow systems. The basic process in the SBR involves an alternative sequence-like fill, react, settle, and draw phase in a single reactor. The reactor operates in batch mode, with aeration and sludge settlement in the same tank. The treatment can be performed in a single basin or multiple basins, allowing for a smaller size of the tank and spaces. The system can simultaneously treat organic carbon, ammonia, nitrogen, and phosphorous in a single tank with fill and draw mode by changing the operating condition from aerobic to anoxic and vice versa [7,8].

According to a review of previous studies, no study has been conducted on the sequencing nitrification–autotrophic denitrification process using $S^0$ as an electron donor. Aspects such as the nitrogen loading rate (NLR) and kinetics are unknown. Therefore, the aim of this research was to develop a nitrification–autotrophic denitrification process using $S^0$ as an electron donor in a single SBR, exploring the effect of an increasing NLR on the performance process. In addition, the kinetic parameters of the process in the SNAD-SBR system through nitrification and denitrification bio-experiments were also assessed.

## 2. Materials and Methods

### 2.1. Inoculum, Synthetic Wastewater, and Experimental Setup

Two types of inoculum were used: first, anaerobic sludge from a clogged anaerobic lagoon treating pig slurry was used as a source of denitrifying biomass; the volatile suspended solid (VSS) concentration of this sludge was 53.47 (g VSS/L). Second, aerobic sludge taken from the covered lagoon of a chicken slaughter plant was used as a nitrifying inoculum, which had a VSS concentration of 1.8 (g VSS/L).

The composition of synthetic wastewater was developed based on the stoichiometry used by the authors Wiesmann [9] and Darbi and Viraraghavan [10]. In addition, traces of other elements (micro-nutrients) were added to the denitrifying feed based on Koenig and Liu's experiments [11], and other elements were utilized for the nitrifying feed [12]. A phosphate buffer solution ($K_2HPO_4$-$KH_2PO_4$) was used for pH control. For each synthetic

substrate, hydrochloric acid (HCl) or sodium hydroxide (NaOH) was added so that the pH of the feed was between 7.0 and 7.3 at the beginning of the cycle.

A transparent acrylic reactor with a cover and mechanical stirrer was used for the laboratory experiments. This reactor had a maximum effective volume of 3 L. The reactor was designed (Figure 1) with three inlet ducts (1, 2, and 3), an outlet duct (5), and a duct that allowed for sampling and the measurement of the pH inside the reactor (4). It also had an aeration system consisting of three air diffusers (7, 8, and 9) and a mechanical stirrer (6). The reactor was inoculated with 1 L of a mixture of aerobic/anaerobic sludge at a rate of 30/70, an estimate based on the research by Guerrero et al. [13]. Finally, there was an initial biomass concentration of approximately 9 [g/L] (it should be noted that this biomass contained microorganisms of all types, including heterotrophs, which would decrease over time due to the lack of organic substrate).

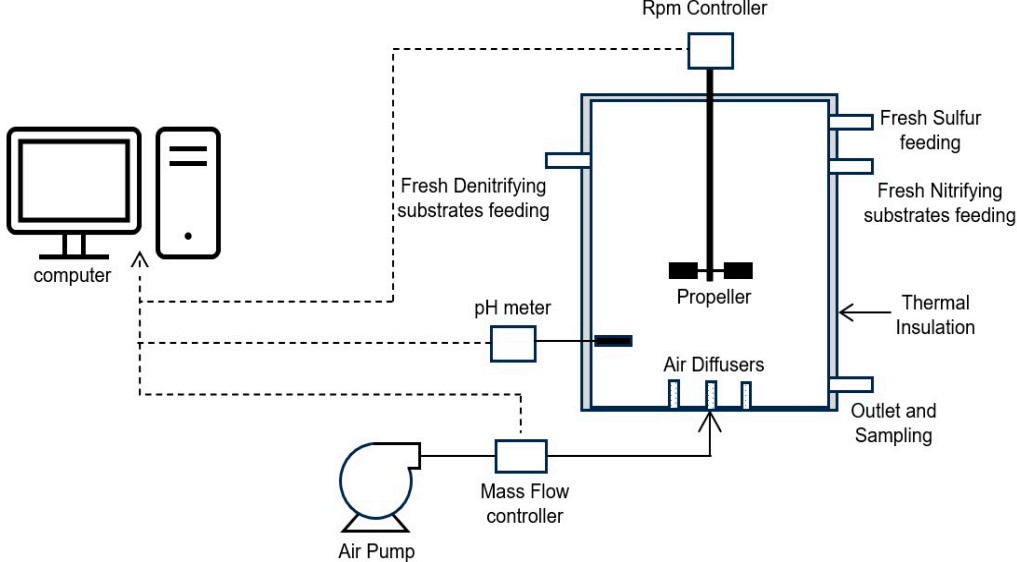

**Figure 1.** Diagram of the sequential batch reactor (SBR).

The substrates were supplied to the reactor by peristaltic pumps for the nitrifying and denitrifying substrates, while elemental sulfur ($S^0$), used in denitrification, was added at the beginning of each cycle. A flow of 7.8 (mL/min) was set for the nitrifying feed and 3.3 (mL/min) for the denitrifying feed. Three diffusers were installed at the bottom of the reactor to supply oxygen inside it, using aquarium air compressors, which achieved an airflow of 2.96 (L/min). Stirring was provided by a rotating blade installed at the top of the reactor (10) and maintained at 100 rpm.

Timers connected to each of the pumps and aerators controlled the execution of the different phases within the cycle. The timers activated each period according to each phase of the SBR cycle. The reactor was mounted inside a thermal cage using a pen and a metal structure. Inside the insulating chamber was a Pt-100, connected to a controller set at $31 \pm 1\,°C$.

### 2.2. Experimental Design

The reactor's operation was designed based on a cycle with a hydraulic retention time (HRT) of 72 h distributed in 3 feeds (one cycle was composed of 3 feeds, i.e., three sub-cycles). The cycle ended with the sedimentation of the treated effluent (45 min) and subsequent discharge (15 min). Each sub-cycle began with a feed corresponding to one-third of the total influent that must enter the reactor and continued with aerobic and anoxic reaction stages, as shown in Table 1. The times for each cycle stage were defined based on lab experiences, considering that the anoxic reaction of denitrification is the limiting

phase within the SNAD cycle. The feeding, sedimentation, and discharge stages constitute periods without air injection, so they are anoxic and contribute to denitrification.

**Table 1.** Phases of each sub-cycle.

| Phase | Start Time, h | End Time, h | Duration, h |
|-------|---------------|-------------|-------------|
| Anoxic feed | 0 | 1 | 1 |
| Anoxic reaction | 1 | 9 | 8 |
| Aerobic reaction | 9 | 17 | 8 |
| Anoxic reaction | 17 | 23 | 6 |
| Aerobic reaction | 23 | 24 | 1 |

It was decided during start-up to progressively increase the nitrogen loading rate (NLR) by 20% over the substrate operating conditions of the previous cycle. The increase in NLR was applied when the following stability criteria were met: 1—removal of more than 50% was achieved concerning the amount of nitrogen that entered the feed plus nitrogen that was in the remaining volume after the discharge of the previous cycle; 2—throughout the previous cycle, the pH was always maintained between 7.0 and 7.6, such that the missing nitrogen in the mass balance was a product of the denitrifying reaction that released atmospheric nitrogen and not of ammonia stripping; 3—the preceding criteria had to be fulfilled for at least three cycles.

There were two operational stages: start-up and operation. Table 2 shows all the initial TAN concentrations and NLRs. The criteria used to establish which stage of the experimental planning was being operated were the behavior of microbial growth and the reactor performance related to the adaptation of the biomass in the different feeding conditions.

**Table 2.** Different nitrogen load rates applied in the study.

| Time, d | TAN, mg/L | NLR, mg/L d |
|---------|-----------|-------------|
| 0–18 | 177.33 | 59.11 |
| 21–27 | 212.8 | 70.9333333 |
| 28–48 | 255.36 | 85.12 |
| 48–75 | 212.8 | 70.9333333 |
| 75–108 | 234.61 | 78.2033333 |
| 109–154 | 223.44 | 74.48 |

The start-up consisted of achieving the adaptation of the microorganisms that carried out the simultaneous processes [14]. Their substrate consumption rates were used to evaluate the microorganism activity based on the removal levels achieved by the process. Then, the feeding stage continued with a fixed composition. During this entire operation, the carbon to nitrogen ratio (C/N) remained constant, both for the nitrifying and denitrifying feeds, concerning inorganic carbon sources ($HCO_3^-$).

For all conditions, the following compounds were measured at the end of each assay: total ammonia nitrogen (TAN), nitrite, nitrate, sulfate, and total suspended solids (TSS).

For nitrification, the efficiency of the process ($\eta_n$) in the cycle is defined as a function of the amount of ammonium remaining in the reactor and in the previously settled discharge (Equation (1)):

$$\eta_n = \frac{m_{i,ammonium} - m_{f,ammonium}}{m_{i,ammonium}} \tag{1}$$

where $m_{i,ammonium}$ is the initial amount of ammonium in the cycle, measured as milligrams of nitrogen per liter (mg N/L), and $m_{f,ammonium}$ (mg N/L) is the final amount of ammonium in the reactor.

The total nitrogen removal at the end of the cycle will depend on how efficient the nitrifying and denitrifying microorganisms have been. The initial amount of nitrite and nitrate, the available oxidized nitrogen, is a function of how much ammonium is oxidized in the nitrification stage. The denitrification efficiency ($\eta_d$) can be written as in Equations (2) and (3):

$$\eta_d = \frac{m_{r,nitrogen}}{m_{ao,nitrogen}} \tag{2}$$

$$\eta_d = \frac{m_{ao,nitrogen} - m_{fo,nitrogen}}{m_{ao,nitrogen}} \tag{3}$$

where $m_{r,nitrogen}$ (mg N/L) is the removed nitrogen, $m_{ao,nitrogen}$ (mg N/L) is the available oxidized nitrogen, and $m_{fo,nitrogen}$ (mg N/L) is the final oxidized nitrogen, which is the amount of nitrite and nitrate at the end of the cycle. The initial amount of nitrite and nitrate is given by Equation (4):

$$m_{ao,nitrogen} = m_{i,ammonium} - m_{f,ammonium} \tag{4}$$

The initial amount of ammonium ($m_{i,ammonium}$) considers the ammonium added in the feed and the remainder in the previous stage.

### 2.3. Kinetic Assays

#### 2.3.1. Aerobic Oxidation Tests of Elemental Sulfur

These tests are designed to determine whether the oxidation of sulfur is the product of denitrification alone or whether injecting oxygen during nitrification also causes the production of sulfates. To do this, an experiment was carried out in which 2 g of sulfur beads was mashed and then brought to a capacity of 500 mL in a beaker. The elemental sulfur tablets had an average ratio between 0.2 and 1 mm, with a concentration of approximately 99%. Then, the same aquarium compressor system was used to aerate the vessel with a flow rate of 1.96 (L/min), maintaining the system temperature at 31 °C. The dissolved oxygen and sulfate concentrations were measured over time to determine if the oxidation of the sulfur produced by the oxygen injected into the system occurred.

#### 2.3.2. Nitrification Tests

Four bioassays (nitrification) were performed at different initial concentrations to determine the kinetics of the nitrifying microorganisms. To carry out the experiments, 60 mL was taken from the reactor (with a known concentration of VSS), which was graduated in a 600 mL beaker. Then, a known amount of ammonia was inserted, and the experiment began.

During the sweep, the sample reactors were maintained at the original SNAD-SBR system's operating temperature of 31 °C and with a constant air flow of 1.96 (L/min). The concentrations of all nitrogenous compounds (ammonium, nitrite, and nitrate) and the concentration of dissolved oxygen at different times were measured.

The four nitrification experiments were carried out with different initial concentrations of ammonia. One was performed at a high ammonium concentration of 400.8 (mg N/L), another at a low ammonium concentration of 11.1 (mg N/L), and the other two bio-experiments were performed at average concentrations similar to those used by Beristain-Cardoso et al. [15] of 50.0 and 53.9 (mg $NH_4^+$/L).

#### 2.3.3. Denitrification Tests

Two denitrification experiments were carried out. For each experiment, a 50 mL volume was taken from the SNAD-SBR reactor and diluted 1:3 in beakers. Then, an initial amount of $KNO_3$, $NH_4^+$, and $NaHCO_3$ in stoichiometric quantities, 1 g of elemental sulfur

in tablet form, and $K_2HPO_4/KH_2PO_4$ (as a buffer solution) were added to each beaker. The elemental sulfur tablets had an average ratio between 8 and 10 mm and a 1 mm thickness, with a concentration of approximately 99%.

The beakers were sealed to maintain the anoxic environment and shaken at 70 rpm. Then, as for the nitrification experiments, periodic samples were taken. In this case, the pH was measured to keep it controlled according to the same operating conditions of the SNAD-SBR system. In addition, the concentration of nitrates and sulfates was measured for each sample to empirically demonstrate the relationship of consumption and production between these two compounds.

### 2.4. Kinetic Model Development

The SNAD process is carried out by two parallel reactions: nitrification and denitrification. The respective kinetics are given by the microbial groups' characteristics, the growth rates, and the inhibition elements.

Nitrification is an aerobic respiratory process carried out by two groups of bacteria: ammonia oxidants and nitrite oxidants. Low growth rates and low yields characterize autotrophic ammonia-oxidizing bacteria. Therefore, nitritation is generally considered the limiting step in nitrification [16].

Haldane-type kinetics provide the most straightforward description of substrate inhibition. Monod-type kinetics are also considered to compare and establish whether there are significant differences. We consider the Haldane-type or Monod-type kinetics for nitrifying biomasses, i.e., AOB and NOB. The inhibition of a substrate according to Haldane kinetics, coupled with the switch function, is described by Equation (5) [9]:

$$\mu_{AOB,NOB} = \mu_{\max_{AOB,NOB}} \frac{C(NH_4^+, NO_2^-)}{K_{s_{AOB,NOB}} + C(NH_4^+, NO_2^-) + \frac{C(NH_4^+, NO_2^-)^2}{K_{i_{AOB,NOB}}}} \frac{C'}{K' + C'} \quad (5)$$

The switch function corresponds to the term $\frac{C'}{K'+C'}$, which describes the availability of oxygen, where $K'$ is different if it corresponds to the oxidation of ammonium or nitrite. For high DO concentrations ($C' \gg K'$) and for a high inhibition constant or low substrate concentration ($K_i \gg NH_4^+$), the equation takes the typical form of Monod-type kinetics (Equation (6)) [9]:

$$\mu_{AOB,NOB} = \mu_{\max_{AOB,NOB}} \frac{C(NH_4^+, NO_2^-)}{K_{s_{AOB,NOB}} + C(NH_4^+, NO_2^-)} \quad (6)$$

For denitrification, the electron donor is elemental sulfur, and its availability in the dissolved phase is essential since elemental sulfur is insoluble in polar solvents. For this, the microorganisms within the reactor must manage to generate a biofilm on the sulfur beads to subsequently destroy the structure and allow the sulfur to be available for autotrophic denitrification [17]. In this sense, using a sulfur-filled bed reactor, Koenig and Liu [11] showed that a medium-order kinetic model can describe the autotrophic denitrification by the biofilm in the reactor. This is because the penetration of the substrate into the pores of the biofilm is less than fully effective, and a zero-order reaction (Equation (7)) in the biofilm is converted into a medium-order reaction (Equation (8)) on the biofilm surface [10].

$$-\frac{dC}{dt} = r_v = k_{1/2\,v}C^{1/2} \quad (7)$$

$$C_e^{1/2} = C_0^{1/2} - \frac{1}{2}k_{1/2\,v}t \quad (8)$$

However, it is also possible that the reaction is first-order (Equation (9)), depending on the behavior of the biofilm, or zero-order (Equation (10)), since the substrate concentrations are very high.

$$\frac{dC\left(NO_3^-\right)}{dt} = -k_1 \cdot C\left(NO_3^-\right) \tag{9}$$

$$\frac{dC\left(NO_3^-\right)}{dt} = -k_0 \tag{10}$$

The mathematical problem was solved using a fourth-order numerical Runge–Kutta method with an integration step of 0.5/1000 (day).

The models selected to represent the kinetics of the AOB microorganisms at the nitrification stage were those of Monod and Haldane. The simulation was contrasted with the experimental data. Then, using the Excel tool Solver, the parameters were adjusted using the numerical analysis technique of the mathematical optimization of least squares, defining $F_0$ as the function to minimize (Equation (11)):

$$F_0 = \sum_{i=1}^{n} \left(S_{i,experiment} - S_{i,modelled}\right)^2 \tag{11}$$

The model with the least error and the highest coefficient of determination will best represent denitrifying microorganisms.

### 2.5. Chemical Analysis

Nitrogenous compounds and sulfates from samples were measured using a spectrophotometer. All methods were extracted from APHA [18]. For nitrates, a measurement at 220 nm quickly determines nitrate since organic matter does not absorb to such a magnitude, and nitrate does not do so at 275 nm. A second measurement at 275 nm corrects the nitrate value (Method 4500 $NO_3^-$ B Ultraviolet Spectrophotometric Screening Method). For nitrites, the method is based on forming a reddish-purple azo dye. The color system obeys the Lambert–Beer law at 543 nm (Method 4500 $NO_2^-$ Colorimetric Method). For ammonia, the method is based on the formation of an intense blue compound, indophenol, which is proportional to the amount of ammonia present in the sample. The spectrophotometric measurement is taken at 635 nm (Method 4500 $NH_3^-$ Phenate Method (F)).

The sulfate ion precipitates in an acid medium with barium chloride, forming barium sulfate crystals of uniform size. Its amount is proportional to the concentration of sulfates present in the sample and the light absorbance of the suspension. It can be measured spectrophotometrically at 420 nm (Method 4500 $SO_4^{2-}$ (E) Turbidimetric Method).

The total suspended solids were determined using the APHA [18] (Method 2540).

## 3. Results and Discussion

### 3.1. Performance of Nitrification–Autotrophic Denitrification in an SBR Using $S^0$ as an Electron Donor

The profiles of the nitrogen concentrations (TAN, $NO_3^-$, and $NO_2^-$) and N-removal efficiency are shown in Figure 2. The initial amount of nitrite and nitrate present is a function of the ammonia oxidized during nitrification. It should consider the residual amount remaining in the reactor after sedimentation and discharge. The quantity of nitrogen at the beginning of the cycle is not only a function of the load set in the feed but also depends on the ammonia, nitrite, and nitrate remaining at the bottom (0.2 L), which, in turn, depends on the efficiency of the previous cycle. This is why the amount of nitrogen added to the system at the beginning of each cycle cannot be fully controlled; where the nitrogen concentration performance throughout the experiment is shown in Figure 2A, the average efficiency of the nitrification process (aerobic process) during the operation of the reactor was 98.63% and relatively stable.

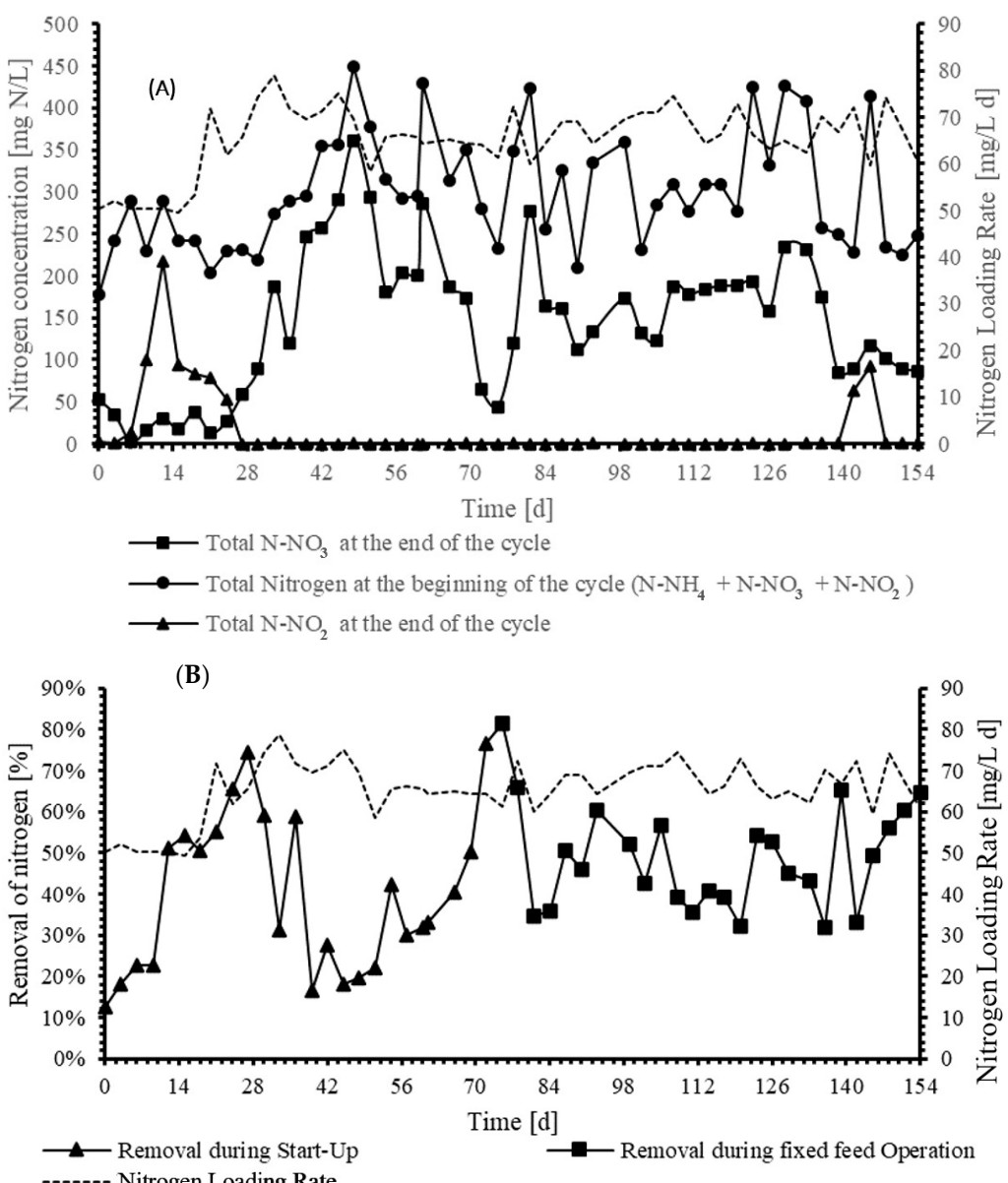

**Figure 2.** Profiles of nitrogen during SBR operation. (**A**) Concentrations of different N compounds at different nitrogen loading rates (NLR); (**B**) nitrogen removal at different NLRs.

In comparison, the average denitrification efficiency throughout the operation was 44.9%. This means that denitrification using $S^0$ was the reaction that controlled all of the N removal. The performance of the aerobic process, considering the high efficiencies achieved, can be explained by the formation of microenvironments within the gradient of the biofilm formed on the surfaces of the sulfur beads, where anoxic/aerobic conditions exist [19]. This promotes oxidative metabolism between the microbial populations involved in the aerobic process of the SNAD system [20], along with a higher concentration of microorganisms due to sedimentation and cellular retention, resulting in a longer retention time.

The denitrification process (anaerobic) depends on the preceding metabolite (nitrification: oxidation of ammonium), and, with a 10% increase in load in each cycle, low average efficiencies were obtained.

Moreover, Figure 2B shows the overall behavior of the simultaneous process (all cycles) from the reactor start-up, where nitrite accumulation generated inhibitory levels at some points, and a noticeable decrease in yields coincided with the increase in the NLR. Only one study [21] has proposed SNAD using $S^0$ as an electron donor in the literature. The

authors indicated that a fluidized bed reactor for SNAD could not perform nitrification because of the presence of sulfide, resulting in very low efficiency in TAN oxidation.

As mentioned earlier, autotrophic denitrification is the reaction that limits the nitrogen removal efficiency. This behavior can be related to the particle size used in our work. It is known that $S^0$ has poor water solubility, which restricts the mass transfer between the solid and liquid phases and produces a lower reaction rate [22]. The S0 particle sizes greatly influence the mass transfer rate, which, in turn, affects the $NO_3^-$-N removal efficiency [23,24].

Although the efficiency is low compared to other works [8,24], the simultaneous process using elemental sulfur as an electron donor can eliminate the nitrogen, and the development of a kinetic model and the obtention of the kinetic parameters are necessary.

### 3.2. Calibration of the Kinetic Model

For nitrification, a Monod-type kinetic model was selected for AOB microorganisms and a Haldane-type kinetic model for NOB microorganisms. The values of the coefficients of determination and the coefficients of determination adjusted (or corrected) for the adjustments based on the selected models are presented in Table 3.

**Table 3.** Parameters for Monod kinetic model applied to nitrification process.

| Monod kinetic model | | | | |
|---|---|---|---|---|
| Type of microorganism | $\mu_{max}$ [d$^{-1}$] | $K_s$ [mg/L] | $K_i$ [mg/L] | $F_0$ |
| AOB | 0.791 | 74.18 | --- | 517.4 |
| NOB | 0.116 | 8.87 | --- | 0.274 |
| Haldane kinetic model | | | | |
| Type of microorganism | $\mu_{max}$ [d$^{-1}$] | $K_s$ [mg/L] | $K_i$ [mg/L] | $F_0$ |
| AOB | 0.807 | 76.27 | 18177.30 | 518.48 |
| NOB | 0.822 | 86.06 | 1.77 | 0.127 |

The reason that the kinetics that best describes the behavior of AOB microorganisms is the Monod-type model is that the inhibition effect byproduct is suppressed by using a buffer, which regulates the concentration of protons in the medium and, therefore, does not allow for sudden fluctuations in the pH. However, the Monod kinetic model is not a good fit. Furthermore, the parameters obtained are far from those in the literature [10,11,25], where the values for $\mu_{maxNOB}$ are always postulated to be higher than those for $\mu_{maxAOB}$. This does not happen when fitting the data according to Monod's kinetics, but it does when fitting them according to Haldane's kinetics.

The results of the nitrification tests and the comparison of the experimental data with the theoretical ones predicted by the models assayed are shown in Figure 3.

The maximum rates $\mu_{max}$ obtained using the best adjustments (including their standard deviations) are compared to the values reported in the literature [9].

In the case of denitrification, zero- and first-order reaction kinetics were also used. The parameter values for each reaction order are shown below, and the determination coefficient ($R^2$) and the sum of the squared errors ($F_0$) are also included. The results of the adjustment for denitrification are presented in Table 4.

**Table 4.** Results of the adjustments for denitrification.

| Kinetic Model Order | Value of the Kinetic Constant | $F_0$ | $R^2$ Experiment 1 | $R^2$ Experiment 2 |
|---|---|---|---|---|
| Order 1 | $k_1 = 0.0112$ h$^{-1}$ | 1472.2 | 0.956 | 0.887 |
| Order 0 | $k_0 = 3.5575$ h$^{-1}$ | 1125.6 | 0.934 | 0.896 |
| Order ½ | $k_{1/2\,v} = 0.2054$ mg$^{1/2}$/L$^{1/2}$/h | 478.6 | 0.957 | 0.979 |

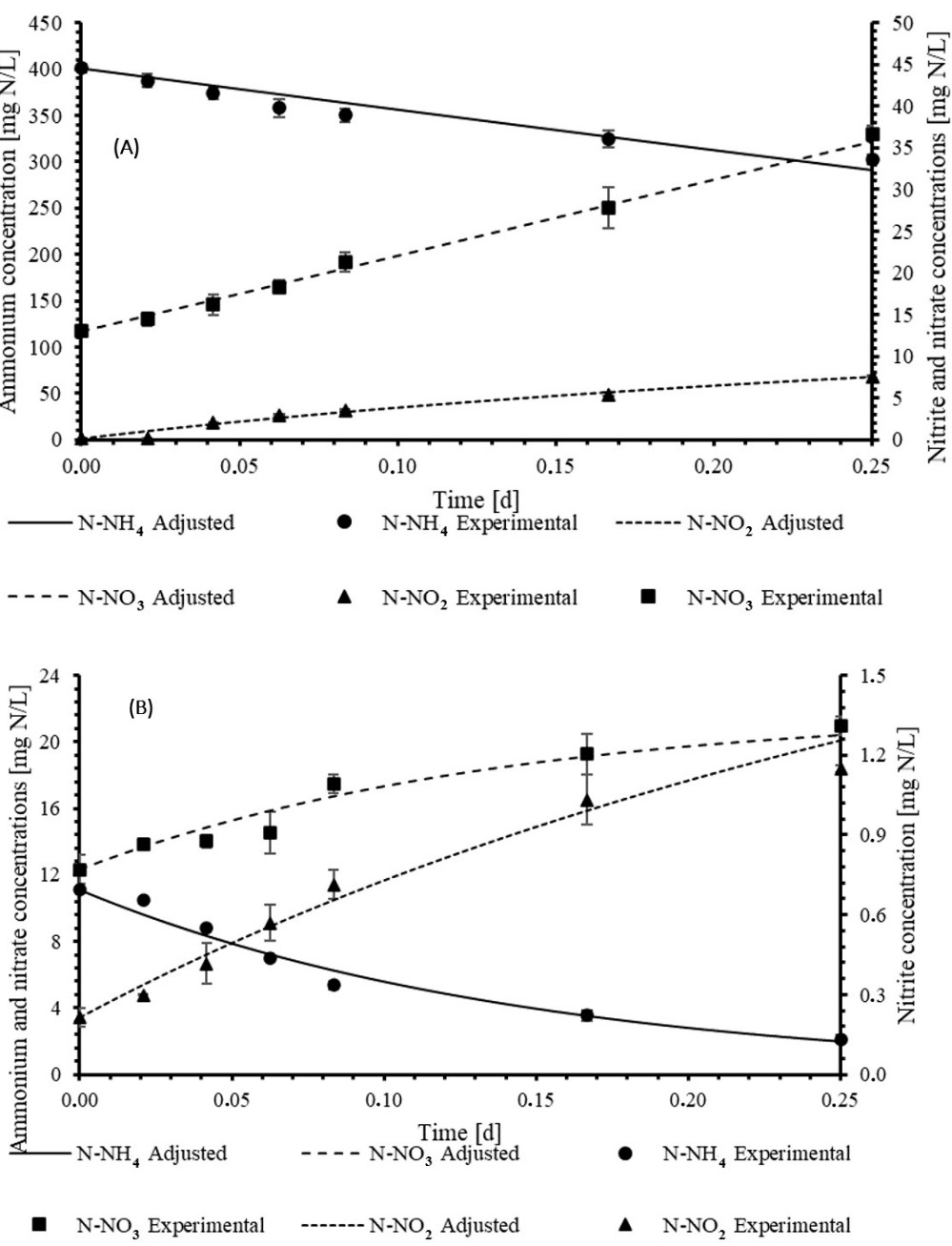

**Figure 3.** Profiles of kinetic calibration. (**A**) Profiles at high TAN concentration; (**B**) profiles at low TAN concentration.

Therefore, it is concluded that the denitrification kinetics is of order ½, consistent with the results reported by Koenig and Liu [11], and with the good adjustment of the parameters. The results of the adjustment are shown in Figure 4. It is essential to indicate that $S^0$ was not oxidized to $SO_4^-$ by chemical reaction, as is shown in the Supplementary Data (see Figure S1).

There is no indication of an increase in the sulfate concentration over time. Therefore, the interaction is ruled out, and it is validated that the generation of sulfates is only a product of denitrification.

As the nitrate is consumed, the sulfate concentration increases. For the denitrification kinetics, the model is adjusted to a nitrate consumption rate proportional to the root of its concentration. It is taken as a valid assumption that denitrification occurs on the surface of

the sulfur pellet. Therefore, a biofilm is generated, which, in two stages, uses sulfur as an electron donor to carry out nitrogen removal [4].

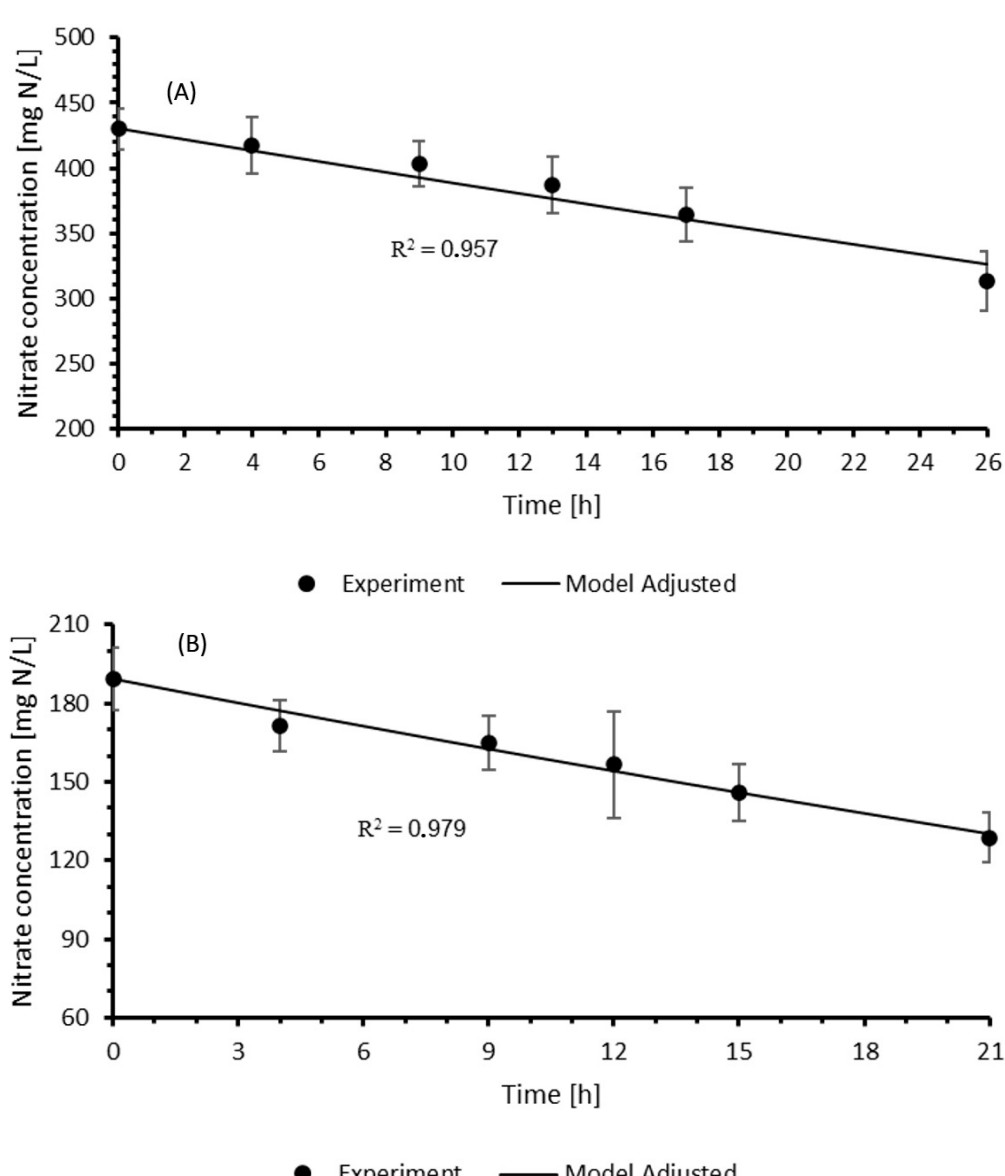

**Figure 4.** Nitrate profiles for denitrification kinetic calibration. (**A**) Profiles at high $NO_3$-concentration; (**B**) Profiles at low $NO_3$-concentration.

### 3.3. Validation of the Kinetic Model Proposed Using SBR Data

To validate the model, which has already been determined to be a good fit, a comparison of the theoretical versus experimental behavior is conducted for selected cycles, as presented in Table 5. It is important to note that for the amount of ammonia nitrogen entering, the nitrogen quantity remaining after discharge (dead volume remaining after discharge) must be considered. A simulation is performed to observe the theoretical behavior of the SNAD-SBR reactor for cycles 30, 34, 36, 40, 42, and 45, respectively.

Concerning the relationship between the kinetic models and the operational behavior of the reactor, in two of the three cases, the experimental efficiency is higher than the theoretical one. One efficiency is higher than the other because the nitrate concentration in the remaining volume of the previous cycle is higher (in the case of greater efficiency). Therefore, the system has more nitrogen in the form of nitrate to remove.

**Table 5.** Comparison of efficiency between simulated cycle and experimental data.

| Cycle | Nitrogen Introduced in the Cycle [mg] | Nitrate Theoretically Removed [mg] | Cycle's Theoretical Efficiency [%] | Cycle's Experimental Efficiency [%] |
|---|---|---|---|---|
| 30 | 237.58 | 104.21 | 43.86 | 50.54 $\pm$ 7.54 |
| 34 | 239.57 | 105.23 | 43.92 | 42.69 $\pm$ 4.60 |
| 36 | 246.31 | 104.19 | 42.30 | 39.12 $\pm$ 6.62 |
| 40 | 262.50 | 108.03 | 41.15 | 39.23 $\pm$ 3.31 |
| 42 | 218.21 | 103.07 | 47.23 | 52.67 $\pm$ 9.73 |
| 45 | 233.63 | 72.28 | 30.94 | 31.90 $\pm$ 5.12 |

Finally, the limiting stage is denitrification, where nitrate is consumed much more slowly than the rate at which ammonium and nitrite oxidize. This is consistent with the kinetic models validated [26] for similar systems without considering elemental sulfur.

**4. Conclusions**

- Simultaneous nitrification–autotrophic denitrification using $S^0$ as an electron donor was performed successfully, with total N-removal efficiency closer to 50% (average). Autotrophic denitrification controls the whole process. Elemental sulfur was only oxidized by the denitrifying biomass, without oxidation by chemical reaction with dissolved oxygen.
- A simple kinetic model was proposed, calibrated, and validated. Nitrification could be modeled by the Haldane kinetics, while autotrophic denitrification obeys the ½ order kinetics. The parameters obtained in both processes are within the range of values presented in the literature.

**Supplementary Materials:** The following supporting information can be downloaded at: https://www.mdpi.com/article/10.3390/su16104269/s1, Figure S1: Profile of sulfate and dissolved oxygen concentrations in the aerobic abiotic experiment using only $S^0$.

**Author Contributions:** Conceptualization, L.G. and C.H.; methodology, L.G. and C.H.; software, M.C.; validation, M.C.; formal analysis, L.G., C.H., C.D.S., M.C., and A.B.; investigation, M.C. and C.D.S.; resources, M.C.; writing—original draft preparation, M.C. and C.D.S.; writing—review and editing, A.B. and C.H.; supervision, L.G. and C.H.; project administration, L.G.; funding acquisition, L.G. All authors have read and agreed to the published version of the manuscript.

**Funding:** This research was funded by Fondecyt-ANID Chilean Project N° 1201258 and Multidisciplinary-USM Project PI-M-2022-07.

**Institutional Review Board Statement:** Not applicable.

**Informed Consent Statement:** Not applicable.

**Data Availability Statement:** The data supporting the results can be obtained by emailing andrea.barahona@usm.cl.

**Acknowledgments:** The authors appreciate the collaboration of their friend Silvio Montalvo (RIP), whom inspired them to continue research.

**Conflicts of Interest:** The authors declare no conflicts of interest.

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
