# Peer review of "Nitrification–Autotrophic Denitrification Using Elemental Sulfur as an Electron Donor in a Sequencing Batch Reactor (SBR): Performance and Kinetic Analysis"

_sustainability, doi:10.3390/su16104269_

Round 1

Reviewer 1 Report

Comments and Suggestions for Authors

This study developed a sequential nitrification-autotrophic denitrification process using S0 as an electron donor in a single SBR. The effect of increasing NLR over the performance process was explored. In addition, the kinetic parameters of the process in the SND-SBR system by means of nitrification and denitrification bio-experiments were also assessed. After addressing the following questions, I believe this study will be of interests for the readers of Sustainability. However, there exists some flaws resulting in current version need further modification:

1.It is mentioned in the paper that the use of S0 as an electron donor is the first time in a sequential nitrification denitrification reactor, but there is not enough detailed information about the details of the S0, such as the type of S0 and the S0 particle size, which significantly affects the rate of denitrification.

2.It is recommended that the operating cycles and loads of the reactors are summarized in a table, especially the specific information on the nitrifying and denitrifying loading of the reactors, so that the reader can go and get the information quickly.

3. How is the nitrogen loading speed (20%) enhancement criterion set for the start-up phase? Why is the stability criterion set only at a removal rate of more than 50%?

4. It is mentioned that the carbon and nitrogen ratio (C/N) remains constant throughout the operation, but no specific information on carbon and nitrogen ratios is given in the paper, and the corresponding COD data do not appear in the graphs.

5. The authors could perform microbial community tests on S0 particle surface biofilms to further determine the presence of S0-oxidizing bacteria. Because there may be the presence of S0-reducing bacteria competing with S0-oxidizing bacteria for the limited number of active sites on the surface of S0 (electron donor or acceptor)

6.Why not choose the same Monod-type model to describe the behavior of NOB microorganisms?

7. The conclusion of the paper mentions that S0 is only oxidized by denitrifying biomass and does not react chemically with dissolved oxygen. However, no relevant experimental data were seen.

8. Penetration of oxygen is one of the key factors influencing denitrification granules or biofilms in general. Since the formation of anoxic zone is strictly related to the denitrification process. Therefore, the aeration method is crucial and the design of the aeration method and the detection of dissolved oxygen can be studied in depth.

Comments on the Quality of English Language

Language can be improved.

Author Response

We deeply appreciate the reviewers' efforts. We have carefully considered all their suggestions and incorporated their comments into the revised manuscript, which is painted in yellow.

Reviewer 2 Report

Comments and Suggestions for Authors

This research aimed to investigate the performance and kinetics of a simultaneous nitrification-autotrophic denitrification process in a lab-scale sequential batch reactor (SND-SBR). It is an interesting paper for the readers/researchers and seems to be important for the topic of nitrogen removal with future applications. The manuscript corresponds to the sustainability journal profile; However, even though the manuscript is quite complete and interesting, it cannot be published until the following suggestions are taken into account:

·      I suggest adding a short two-line introduction in the summary to give some context and not go directly.

·      Please correct the following acronyms: Simultaneous nitrification and autotrophic denitrification (SNAD)Simultaneous nitrification and denitrification (SND). Check when they are correct to put in the manuscript appropriately.

·      Add in the introduction line 38, the meaning of the acronyms AOB (Ammonium Oxidizing Bacteria) and NOB (Nitrite Oxidizing Bacteria). 

·      Add in the introduction line 82, the meaning of the acronym NLR (Nitrogen loading rate).

·      in line 94. "the nitrifying and denitrifying substrates" refers to synthetic water? if this is correct I suggest to change "The composition of synthetic water has been developed....."

·      You must indicate with bullet points "which stability criteria were met"(line 143). This is suggested because it makes the continuous reading of the manuscript in those next lines confusing. 

·      You do not show what parameters you are going to monitor for reactor control. Add pls

·      Add bold the year of the following references: 5, 12, 15, 23 and 24. 

·      You show several quite old references, that part could be improved. 

Comments on the Quality of English Language

Minor editing of English language required

Author Response

(The authors gave the same response as above.)

Reviewer 3 Report

Comments and Suggestions for Authors

The article entitled “Nitrification-autotrophic denitrification using elemental sulfur as an electron donor in a sequencing batch reactor (SBR): Performance and kinetic analysis presents experimental studies on adsorptive removal of PVC microplastic using modified pine bark biochar.  The article is well written, however minor revision is needed before this paper can be considered for the publication.

1.              In Abstract, the mechanism and novelty of study on sulfur on nitrification-autotrophic denitrification process need to be discussed.

2.              In Introduction, the literature is inadequate, and the older references could be replaced with current relevant studies.

3.              Figure 1 needs to be replaced with technical bioreactor diagram.

4.              The experimental results of nitrification and denitrification performance can be compared with literature.

5.     Conclusion is short and needs to be elaborated on nitrification-autotrophic denitrification performance and kinetic models.

6.     The language of the manuscript needs to be corrected in the whole manuscript.

Comments on the Quality of English Language

Language correction required

Author Response

(The authors gave the same response as above.)

Round 2

Reviewer 1 Report

Comments and Suggestions for Authors

No further comment